# Interspecies Interactions of Single- and Mixed-Species Biofilms of *Candida albicans* and *Aggregatibacter actinomycetemcomitans*

**DOI:** 10.3390/biomedicines13081890

**Published:** 2025-08-03

**Authors:** Adèle Huc, Andreia S. Azevedo, José Carlos Andrade, Célia Fortuna Rodrigues

**Affiliations:** 1Department of Pharmaceutical Sciences, University Institute of Health Sciences—CESPU (IUCS-CESPU), 4585-116 Gandra PRD, Portugal; a29927@alunos.cespu.pt; 2LEPABE—Laboratory for Process Engineering, Environment, Biotechnology and Energy, Faculty of Engineering, University of Porto, Rua Dr. Roberto Frias, 4200-465 Porto, Portugal; asazevedo@fe.up.pt; 3ALiCE—Associate Laboratory in Chemical Engineering, Faculty of Engineering, University of Porto, Rua Dr. Roberto Frias, 4200-465 Porto, Portugal; 4Associate Laboratory i4HB—Institute for Health and Bioeconomy, University Institute of Health Sciences—CESPU (IUCS-CESPU), 4585-116 Gandra PRD, Portugal; 5UCIBIO—Applied Molecular Biosciences Unit, Translational Toxicology Research Laboratory, University Institute of Health Sciences (1H-TOXRUN, IUCS-CESPU), 4585-116 Gandra PRD, Portugal

**Keywords:** mixed biofilm, *C. albicans*, *A. actinomycetemcomitans*, antimicrobial resistance

## Abstract

Polymicrobial biofilms involving fungal and bacterial species are increasingly recognized as contributors to persistent infections, particularly in the oral cavity. *Candida albicans* and *Aggregatibacter actinomycetemcomitans* are two commensals that can turn into opportunistic pathogens and are able to form robust biofilms. **Objectives**: This study aimed to assess the interaction dynamics between these two microorganisms and to evaluate their susceptibility to fluconazole and azithromycin in single- and mixed-species forms. **Methods**: Biofilm biomass was quantified using crystal violet assays, while biofilm cell viability was assessed through CFU enumeration (biofilm viability assay). To assess the resistance properties of single versus mixed-species coincubations, we applied the antimicrobial susceptibility test (AST) to each drug, and analysed spatial organization with confocal laser scanning microscopy, using PNA-FISH. **Results**: The results indicated that both species can coexist without significant mutual inhibition. However, a non-reciprocal synergism was also observed, whereby mixed-species biofilm conditions promoted the growth of *A. actinomycetemcomitans*, while *C. albicans* growth remained stable. As expected, antimicrobial tolerance was elevated in mixed cultures, likely due to enhanced extracellular matrix production and potential quorum-sensing interactions, contributing to increased resistance against azithromycin and fluconazole. **Conclusions**: This study provides novel insights into previously rarely explored interactions between *C. albicans* and *A. actinomycetemcomitans*. These findings underscore the importance of investigating interspecies interactions within polymicrobial biofilms, as understanding their mechanisms, such as quorum-sensing molecules and metabolic cooperation, can contribute to improved diagnostics and more effective targeted therapeutic strategies against polymicrobial infections.

## 1. Introduction

Biofilm formation is a key virulence trait in both bacterial and fungal infections, as it contributes to antimicrobial resistance and evasion of hosts’ immune defences as well as facilitating horizontal gene transfer, particularly within multispecies communities [1]. The oral cavity hosts polymicrobial biofilms which enclose diverse bacterial and fungal species engaged in complex interkingdom interactions that can either stabilize the microbial community or promote pathogenic shifts contributing to diseases [2]. A notable example is provided by the emerging evidence of a potential synergistic relationship between *Candida albicans*, an oral opportunistic fungal pathogen [3], and *Aggregatibacter actinomycetemcomitans*, a key bacterial agent in periodontitis [4]. Recent studies suggest a positive correlation between *A. actinomycetemcomitans* (a Gram-negative bacteria) and the presence of *C. albicans* in the saliva of COVID-19 patients, with elevated ACE2 expression potentially contributing to their co-occurence [5]. A separate 72 h incubation assay also confirmed a significant increase in *A. actinomycetemcomitans* cell count in the presence of *C. albicans*, supporting a synergistic relationship hypothesis [6].

Individually, both organisms can cause oral and systemic infections. However, in a mixed biofilm, their coexistence may enhance pathogenicity, biomass robustness, and resistance to standard therapies [7]. *C. albicans* is known for its morphogenetic plasticity, its immune evasion capabilities, and its adaptability in biofilm settings [8]. Conversely, *A. actinomycetemcomitans* exhibits strong adhesion via extracellular matrix protein adhesin A (EmaA) and fimbriae [9] and produces immunomodulatory virulence factors like leukotoxin [10]. In the periodontal microenvironment, *A. actinomycetemcomitans* initiates a supra-physiological immune-inflammatory response, disrupting homeostasis of gingival tissues, periodontal ligament, cementum, and alveolar bone, ultimately leading to periodontal destruction and tooth loss [11]. Emerging research also suggests that, in certain poly-organism biofilms, *C. albicans* may create a microenvironment conducive to bacterial survival [12], while *A. actinomycetemcomitans* influences fungal morphology and contributes to structural integrity [5]. These interkingdom interactions can result in enhanced antimicrobial tolerance and persistence of infection, which has been recorded for other mixed biofilms such as *C. albicans* and *Streptococcus gordonii* [13].

The interaction between *C. albicans* and *A. actinomycetemcomitans* in mixed-species biofilms remains poorly described. This study contributes to filling that gap, particularly regarding their spatial structures, growth dynamics, and antimicrobial susceptibility profiles in single- and mixed-species incubation.

## 2. Materials and Methods

### 2.1. Microbial Strains and Culture Conditions

The reference strains *Candida albicans* SC5315 and *Aggregatibacter actinomycetemcomitans* ANH9381 were acquired from the American Type Culture Collection (Manassas, VA, USA). *C. albicans* SC5314 was grown in Sabouraud Dextrose Agar (SDA) (Merck, Darmstadt, Germany), under aerobic conditions, for 24 h at 37 °C [14]. To determine *A. actinomycetemcomitans* growth, a blood agar (BA, Frilabo, Maia, Portugal) medium was used after three days of incubation at 37 °C with 5% CO_2_ [15]. Microbial cultures were stored with culture media (for bacteria and yeasts) with 15% glycerol at −80 °C.

### 2.2. Antifungal and Antibacterial Drugs

Fluconazole (Flu) and azithromycin (Azm) were purchased in pure forms (Sigma Aldrich, St. Louis, MO, USA). For all drugs, aliquots of 5000 mg/L were prepared using dimethyl sulfoxide (DMSO). The final concentrations used were prepared in RPMI-1640 (Sigma-Aldrich, Roswell Park, St. Louis, MO, USA). The concentrations used were 0.125 mg/L and 8 mg/L for azithromycin [16] and 2–4 mg/L for fluconazole [17,18]. The aliquots were stored at −80 °C.

### 2.3. Single- and Mixed-Microbial Biofilm Standardisation

Standardized cell suspensions of 0.1 McFarland for single-microbial biofilms of *C. albicans* and *A. actinomycetemcomitans*, and 0.4 McFarland for single-microbial biofilm with RPMI-1640 (Sigma Aldrich, St. Louis, MO, USA), were placed in 96-well polystyrene microtiter plates (Orange Scientific, Braine-l’Alleud, Belgium). A volume of 100 μL of each species was inoculated for each of the single-microbial biofilms (2× concentrated), and 50 μL volumes of both species (4× concentrated) were inoculated for the mixed biofilm (final volume of 200 μL per well) [19]. Microplate wells with single- and mixed-microbial suspensions were incubated at 37 °C in a chamber with 5% CO_2_ for 72 h.

### 2.4. Biofilm Viability Assay

Standardised cell suspensions of *C. albicans* (0.1 McFarland) and *A. actinomycetemcomitans* (0.1 McFarland) of 200 μL volume were placed into 96-well polystyrene microtiter plates (as a negative control, 200 μL of RPMI-1640 without organisms was used). For the mixed biofilm, 50 μL of each organism (4× concentrated) was inoculated into single wells to which RPMI-1640 was then added, giving a total volume of 200 μL. Each plate was incubated at 37 °C for 72 h with 5% of CO_2_. After 72 h, the content of each well was removed, and 200 μL of phosphate-buffered saline (PBS, 0.1 *M*, pH = 7.2) was carefully added to remove non-adherent cells. Then, biofilms were scraped from the wells and the suspensions were vigorously vortexed for 2 min to disaggregate cells from the matrix. Serial decimal dilutions in PBS were plated on SDA and incubated for 24 h (for *C. albicans* CFU counts) at 37 °C, then plated in blood agar (for *A. actinomycetemcomitans* CFU counts) at 37 °C with 5% CO_2_. The results were presented as total CFUs per volume (Log_10_ CFU/mL) [19].

### 2.5. Biofilm Biomass Quantification with Crystal Violet

The quantification of total biofilm biomass was achieved by means of crystal violet (CV) staining for each of the control, single-microbial, and mixed-microbial suspensions [14]. As described previously, after biofilm formation (72 h), the medium was aspirated, and non-adherent cells then removed by washing the biofilms with PBS. Next, biofilms were fixed with 200 μL of methanol, which was removed after 15 min of contact. The microtiter plates were allowed to dry at room temperature, and 200 μL of CV (1% *v*/*v*) was added to each well over a 5 min period. The wells were then gently washed twice with sterile, ultra-pure water, and 200 μL of acetic acid (33% *v*/*v*)was then added to release and dissolve the stain. The absorbance of the obtained solution was read in triplicate using a microtiter plate reader (Bio-Tek Synergy HT, Agilent, Santa Clara, CA, USA) at 570 nm. The results were calculated in terms of absorbance per unit area (Abs cm^−2^) and presented in terms of percentage of biomass produced, applying the following formulas:$$\%\;reduction\;of\;single\;Ca=100-\frac{OD\;Mixed}{OD\;Ca\;single}\times 100$$$$\%\;reduction\;of\;single\;Aa=100-\frac{OD\;Mixed}{OD\;Aa\;single}\times 100$$

Note: Ca—*C. albicans*; Aa—*A. actinomycetemcomitans*; OD—optical density.

### 2.6. Antimicrobial Susceptibility Testing (AST)

Antimicrobial susceptibility testing (AST) was performed according to the guidelines of the European Committee on Antimicrobial Susceptibility Testing (EUCAST) [17,18,20]. The inoculum was prepared by suspending five distinct colonies of ≥1 mm diameter, from 24 h cultures, in at least 3 mL of sterile distilled water. Next, the inoculum was suspended by vigorous shaking on a vortex mixer for 15 s. The cell density was then adjusted to the density of a 0.5 McFarland standard and sterile distilled water was added as required, giving a yeast suspension of 1–5 × 10^6^ CFU mL^−1^ (colony-forming units). A working suspension was prepared by a dilution of the standardised suspension in sterile distilled water to yield 1–5 × 10^5^ CFU mL^−1^. The 96-well plate was prepared with 100 μL of cell suspension and 100 μL of each antifungal and antibacterial agent (0.125 mg/L and 8 mg/L for azithromycin [16]; 2 and 4 mg/L for fluconazole [17,18]—2× concentrated) and incubated at 37 °C in a 5% CO_2_ chamber for 18–72 h. Controls without antimicrobial agents were also used (positive control: working solution of cells and RPMI-1640; negative control: sterile distilled water and RPMI-1640). Finally, the results were determined by naked-eye observation [19].

### 2.7. Examination of Spatial Arrangements of Biofilms Using PNA-FISH Combined with Confocal Laser Scanning Microscopy

To examine the spatial structures of biofilms, the PNA-FISH method was used in combination with confocal laser scanning microscopy. A specific 23S rRNA PNA probe developed and optimized by our group was used for *Candida* spp. detection: 5′-Alexa488-OO-CACCCACAAAATCAA-3′ (melting temperature: 75.69 °C; specificity: 96.04%; sensibility: 84.79%). The probe was synthesized (Panagene, Daejoen, Republic of Korea), attached to the Alexa^®^-488 fluorochrome, and tested with *Candida albicans* SC5314. Briefly, single- and mixed-species biofilms were formed on 6-well microtiter plates. The biofilms were washed with saline solution to remove loosely bound cells and then placed in Petri dishes to dry at 60 °C for 15 min. After that, they were fixed with 100% methanol for 20 min, followed by 4% paraformaldehyde for 15 min. They were then air-dried on the bench until fully dry.

After this, the FISH procedure was applied. The biofilms were incubated with PNA Probe at 54 °C for a period of 30 min in the dark. For *A. actinomycetemcomitans* (single- and mixed-species biofilms), the biofilm was overlayed with 100 mg/L of 4′,6-diamidino-2-phenylindole (DAPI) (ThermoFisher Scientific, Waltham, MA, United States) for 10 min at room temperature [21]. Finally, the samples were observed using STELLARIS 5 mounted on a Leica DM6 B upright microscope equipped with a white light laser (WLL) as an excitation light source (from 485 to 685 nm) and 2 internal detection channels equipped with Power HyD S. Image acquisition was performed using a 63× oil-immersion objective (63×/1.4 W) and a 488 nm laser line. Z-stacks with 1 µm Z-steps were collected. All microscope settings were identical among the analysed groups. Zeiss Zen software (version 10) was used for confocal image acquisition and processing.

### 2.8. Statistical Analysis

All experiments were independently repeated at least three times in duplicates. IBM SPSS statistic for Windows, version 30.0 (SPSS Inc., Chicago, IL, USA) software was used for the statistical interpretation of data. Results were compared using a two-sample *t*-test. All tests were performed with a confidence level of 95%.

## 3. Results

The study aimed to evaluate how cell-to-cell interactions between *Candida albicans* and *Aggregatibacter actinomycetemcomitans* affected their baseline viable cell counts when grown as single-species biofilms, compared to when they were co-cultured in mixed-species biofilms over 72 h.

### 3.1. Colony-Forming Unit (CFU) Analysis of Single- and Mixed-Species Biofilms of C. albicans and A. actinomycetemcomitans

The biofilm viability assay addressed the fundamental question of whether these species exhibit mutual inhibition, synergism, or neutral coexistence under standard culture conditions. The quantification of CFUs revealed a non-reciprocal synergistic interaction: while *A. actinomycetemcomitans* exhibited a significant increase in viable biofilm-associated cells under mixed-species conditions (300% increase, ** *p* = 0.018), *C. albicans* showed a 15.13% increase in CFU counts in the presence of *A. actinomycetemcomitans*, although this augmentation was not statistically significant (* *p* = 0.495) (Figure 1). These findings highlight a potential asymmetric benefit favouring *A. actinomycetemcomitans* in the mixed biofilm environment.

### 3.2. Biomass Quantification with Crystal Violet

The impact of mixed-species biofilm formation on total biomass production was evaluated by comparing the relative biomasses of *C. albicans* and *A. actinomycetemcomitans* under single- and mixed-culture conditions, as shown in Table 1. Both showed a statistically significant reduction in biomass under the mixed-biofilm condition. The decrease was more pronounced in the case of *C. albicans*, with a 56.07% reduction (*p* ≤ 0.001) recorded, compared with the 19.16% reduction observed for *A. actinomycetemcomitans* (*p* = 0.048). These findings, when considered alongside the viable cell counts (Figure 1), revealed that *C. albicans* maintained CFU counts despite a significant reduction in biomass, while *A. actinomycetemcomitans* showed a 300% increase in CFU counts despite a reduction in biomass.

### 3.3. Antimicrobial Susceptibility Testing

The macroscopic observations from AST assays (Table 2) suggested distinct inhibitory effects of fluconazole and azithromycin on monomicrobial and mixed cultures. In single-species planktonic cultures, *C. albicans* remained susceptible to fluconazole at both 2 mg/L and 4 mg/L. However, the extended incubation period may have allowed the emergence of tolerant/resistant colonies possibly not representative of the original strain [22]. *A. actinomycetemcomitans* was resistant to azithromycin at 0.125 mg/L but susceptible at 8 mg/L, indicating concentration-dependent sensitivity. In contrast, mixed biofilms consistently displayed intermediate susceptibility to fluconazole (2 mg/L and 4 mg/L) and tolerance/resistance to azithromycin (0.125 mg/L and 8 mg/L), suggesting that co-incubation enhances resistance through synergistic or protective interactions, possibly via altered matrix composition or *quorum-sensing* mechanisms.

### 3.4. Biofilm Structure Analysis

Confocal microscopy analysis of biofilm structure (Figure 2 and Figure 3) provided visual insights into the spatial organization and density of *C. albicans* and *A. actinomycetemcomitans* in mono- and mixed-culture conditions after 72 h. In monocultures (Figure 2a for *C. albicans*, Figure 2b for *A. actinomycetemcomitans*), *C. albicans* and *A. actinomycetemcomitans* formed distinct, dispersed clusters with moderate cell density, stained with A-Alexa Fluor 488 and 4′,6-diamidino-2-phenylindole, respectively. In the mixed biofilm (Figure 3), a high-density network emerged, with *A. actinomycetemcomitans* cells (B) adhering to *C. albicans* hyphae (D), and mixed-species clusters (A) coexisting with individual *C. albicans* cells (C). This structure suggests a cooperative rather than competitive interaction, as both species maintained comparable growth without overt dominance. The presence of hyphae likely facilitates *A. actinomycetemcomitans* adhesion, supporting the observed CFU increase, while the dense matrix may contribute to the reduced biomass and enhanced resistance noted in prior analyses.

## 4. Discussion

Periodontitis is very prevalent in Europe, with a severe form of it being described in 48.5% of untreated individuals [23]. The potential of this chronic inflammatory pathology to lead to systemic complications in cardiovascular diseases and diabetes has been well documented [24,25], immunocompromised patients being its principal targets [26]. Despite this, the staging of disease and its relationship with inflammation, reflecting the immune response, and changes in oral integrity remains complex and not fully understood [27]. The contribution of polymicrobial biofilm dysbiosis has proven significant, as this plays a key role in maintaining microbial imbalance, exacerbating the host immune response, and demonstrating higher levels of resistance to conventional antimicrobial therapies [28,29].

In this study, the conservation of *C. albicans* viable cell counts in the presence of *A. actinomycetemcomitans* contrasts with prior reports suggesting that AI-2, produced by *A. actinomycetemcomitans* via the LuxS pathway, inhibits *C. albicans* hyphae formation [30]. This discrepancy may reflect context-dependent dynamics in our mixed-species biofilm model. For instance, environmental factors, such as iron availability, could limit AI-2’s effective concentration through the influence of genetic expression [31]. Alternatively, *C. albicans* might benefit from the mixed environment, possibly by creating microaerophilic or anaerobic niches through its hyphal structures and metabolic activity, which could enhance *A. actinomycetemcomitans* survival [6] and, conversely, support its own growth rather than inhibit it. Overall, these results suggest that mixed-species biofilm formation promotes *A. actinomycetemcomitans* proliferation, while the effect on *C. albicans* reflects a biologically relevant interkingdom interaction that warrants further investigation to elucidate the balance between inhibition and growth promotion.

The reduction in biofilm biomass, as measured by crystal violet, contrasted with the stable CFU counts for *C. albicans* and suggested suppression of hyphal development or a decrease in extracellular matrix (ECM) production in the presence of *A. actinomycetemcomitans.* In fact, the autoinducer-2 (AI-2) secreted by *A. actinomycetemcomitans* is known to inhibit fungal hyphal formation [30,32]. This may explain the finding of a reduction in biomass without any effect on cell viability. In the case of *A. actinomycetemcomitans*, the reduction in biomass may reflect an ecological disadvantage in mixed biofilms, limiting its competitive fitness within complex oral communities, as previously observed with other mixed-species biofilms [33,34]. Biofilm formation by this species is also known to be modulated by factors such as nutrient availability, environmental stress [35], and, particularly, iron limitation [36], conditions which were not specifically controlled or assessed in this study. These unexamined variables might explain the relatively well-preserved biomass under mixed-species settings and highlight the bacterium’s adaptive capacity within polymicrobial communities. Future studies should quantify ECM components and hyphal gene expression in *C. albicans*, and also assess environmental variables, to clarify these mechanisms and their impacts on biofilm dynamics.

Regarding the resistance-enhancing properties of free cells in mixed-species incubation, the expression of hydrolytic enzyme by *C. albicans* has been proven to be exacerbated by the presence of periodontal bacteria [6]. The aspartic proteinases (SAPs), among the most extensively debated of the enzymes produced by *C. albicans*, are of particular interest. Indeed, the SAPs degrade proteins and, to a lesser extent, polysaccharides within the biofilm matrix, altering its composition [37]. This remodelling increases the density of β-1,3-glucan, a polysaccharide known to enhance drug tolerance, as previously reported by our group [19,38,39,40] and also by others [41]. As previously reported, AI-2 *quorum sensing* plays a central role by reducing the hyphal form of *C. albicans*. This yeast has been shown to influence the production of AI-2 by Gram-negative bacteria (such as *A. actinomycetemcomitans*), creating a complex bidirectional communication which is hard to target in therapy [42]. This molecule is also implicated in the signalisation and gene expression of specific virulent components such as the QseBC dual system. It is well documented in the literature that the QseBC can regulate iron acquisition mechanisms which are demonstrably essential for *A. actinomycetemcomitans* survival and virulence [43,44]. A substantial body of existing evidence strongly supports this regulatory relationship being one of the operative mechanisms underlying our observed results.

Biofilm structure analysis, as depicted in Figure 3 (arrow D), revealed that *C. albicans* formed filamentous hyphae during biofilm development to which *A. actinomycetemcomitans* adhered, consistent with prior studies reporting mature mixed-species biofilm by 72 h [6]. However, the biomass which was observed to increase on the previous study, actually decreased in ours,, suggesting that *A. actinomycetemcomitans* may regulate *C. albicans* colonization through *quorum-sensing* mechanisms akin to *Pseudomonas aeruginosa*’s production of 3-oxo-C12-homoserine lactone, which inhibits hyphal formation [45]. Despite the decrease in biomass, the stable CFU counts further support this hypothesis, indicating that *A. actinomycetemcomitans* primarily affects *C. albicans*’s filamentous form rather than its yeast form. Moreover, as reported, *A. actinomycetemcomitans’* global transcriptional regulator H-NS likely modulates these interactions by altering protein expression profiles in neighbouring species within multispecies oral biofilm [46]. These trans-kingdom molecular communications enable both species to adapt to shared niches, influencing community dynamics and survival. 

## 5. Conclusions

This study provides the first comprehensive characterization of mixed biofilms involving *C. albicans* and *A. actinomycetemcomitans*, two clinically important oral pathogens. Our findings revealed a non-reciprocal synergistic relationship in which mixed-species conditions significantly enhanced *A. actinomycetemcomitans* survival (*A. actinomycetemcomitans* alone: ~6 log CFU/mL vs. *A. actinomycetemcomitans* in mixed biofilm: ~7 log CFU/mL), while maintaining *C. albicans* viability despite reductions in biomass production in both species. Critically, we demonstrated that mixed-species biofilms exhibited enhanced tolerance to both fluconazole and azithromycin compared to single-species biofilms, with important implications for clinical treatment strategies.

The coexistence and structural integration of both organisms without competitive exclusion, as confirmed by confocal microscopy, suggests a stable polymicrobial community formation. While our mechanistic hypotheses regarding AI-2 signalling, QseBC regulation, and extracellular matrix modifications are well supported in the literature, direct experimental validation of these pathways in our mixed biofilm model represents a priority for future investigations.

This foundational study establishes essential baseline data for understanding clinically relevant interactions between these species and provides a framework for developing more effective therapeutic approaches against polymicrobial oral infections. Future research incorporating temporal analysis, varying environmental conditions, and transcriptomic/metabolomic approaches will further elucidate the molecular mechanisms underlying these interactions and their therapeutic targets.

## Figures and Tables

**Figure 1 biomedicines-13-01890-f001:**
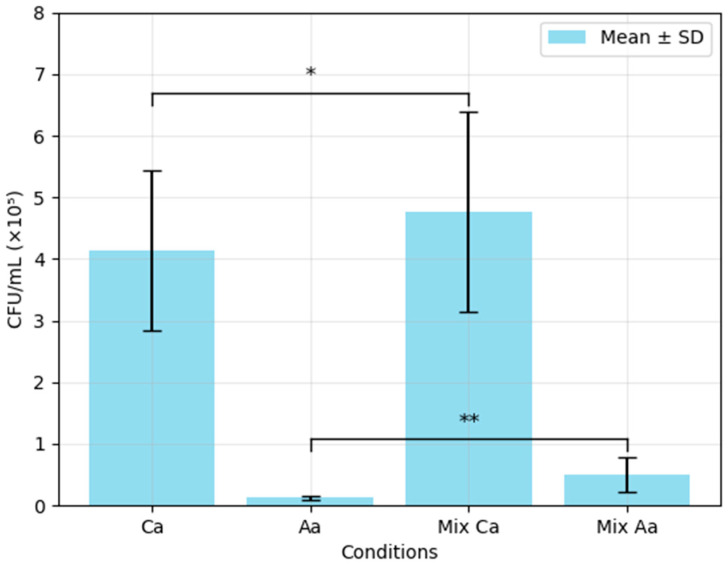
Mean CFU/mL values for *C. albicans* and *A. actinomycetemcomitans* in single- and mixed-species biofilms (biofilm viability assay). Ca—*C. albicans* mono incubation (Ca); Aa—*A. actinomycetemcomitans* mono incubation; Mix Ca—mixed biofilm—Ca CFU count; Mix Aa—mixed biofilm—Aa count. All plates were inoculated for 72 h. Statistical significance was determined using a two-sample test (two-tailed, assuming equal variances): * *p* = 0.495, ** *p* = 0.018, when compared with the respective mono-incubation controls.

**Figure 2 biomedicines-13-01890-f002:**
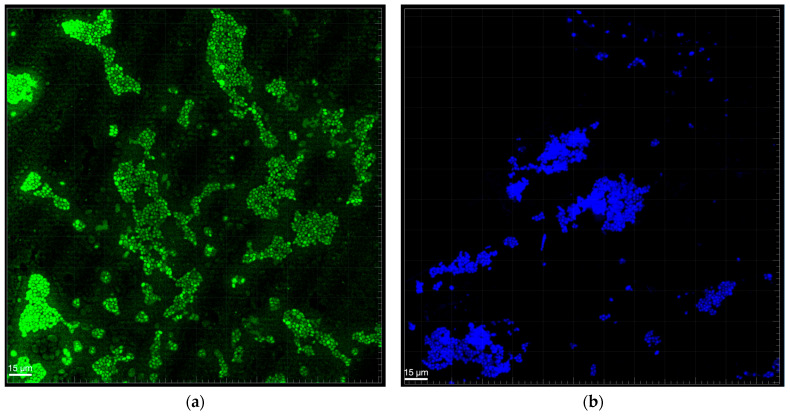
Biofilm of (**a**) *C. albicans* and (**b**) *A. actinomycetemcomitans* in monoculture at 72 h of growth by epifluorescence microscopy confocal (FISH PNA probe with A-Alexa Fluor 488 and 4′,6-diaminidino-2-phenylindole fluorescent stain respectively).

**Figure 3 biomedicines-13-01890-f003:**
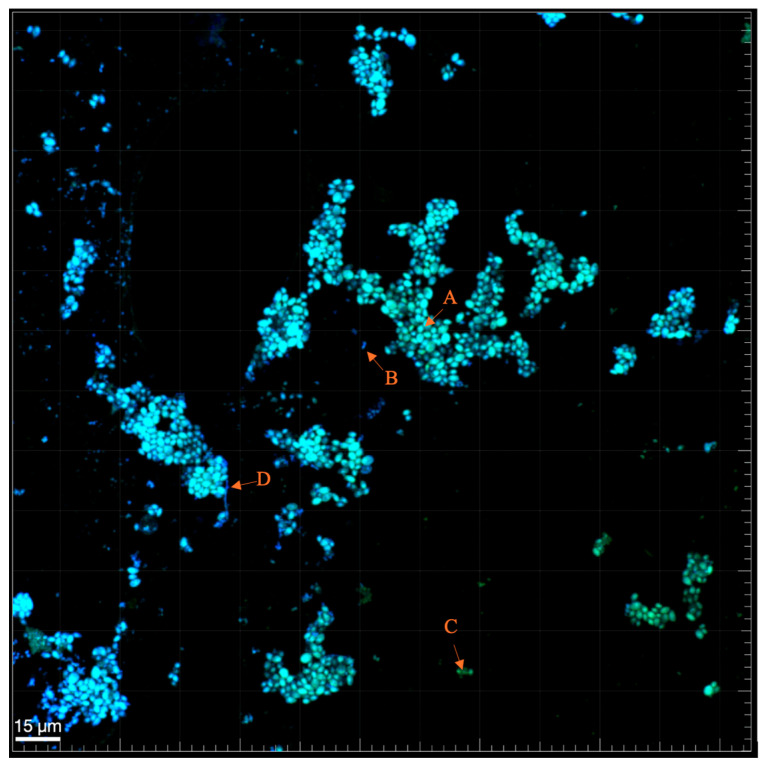
Mixed biofilm of *C. albicans* and *A. actinomycetemcomitans* after 72 h of growth, as imaged by confocal microscopy using a PNA FISH probe with A-Alexa Fluor 488 (single Ca and mixed biofilm cells) and 4′,6-diaminidino-2-phenylindole fluorescent stain (Aa biofilm cells). A: Mixed-species cells; B: *A. actinomycetemcomitans* cells; C: *C. albicans* cells; D: *C. albicans* hyphae.

**Table 1 biomedicines-13-01890-t001:** Results of crystal violet assay of biomass reduction in *C. albicans* and *A. actinomycetemcomitans* after 72 h of single- and mixed-species biofilm growth.

	Optical Density		% Reduction in Single-Species Biofilm Compared with Mixed-Species Biofilm		
Ca	Aa	Mixed			Species
0.724 ± 0.17	0.397 ± 0.12	0.318 ± 0.06	56.07 ± 0.03	*p* ≤ 0.001	*C. albicans*
			19.16 ± 0.01	*p* = 0.048	*A. actinomycetemcomitans*

Note: Ca—*Candida albicans*; Aa—*Aggregatibacter actinomycetemcomitans*.

**Table 2 biomedicines-13-01890-t002:** Differences in susceptibility of planktonic single- vs. mixed-species cultures of *C. albicans* and *A. actinomycetemcomitans* to fluconazole and azithromycin.

Species	Flu 2 mg/L	Flu 4 mg/L	Azm 0.125 mg/L	Azm 8 mg/L
*C. albicans*	Susceptible	Susceptible	-	-
*A. actinomycetemcomitans*	_	_	**Resistant**	Susceptible
Mixed biofilm	**Intermediate**	**Intermediate**	**Resistant**	**Resistant**

Note: Flu—fluconazole; Azm—azithromycin. Bold text indicates intermediate or resistant susceptibility to the drugs.

## Data Availability

The original contributions presented in this study are included in the article. Further inquiries can be directed to the corresponding authors.

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
