# Peer review of "Interspecies Interactions of Single- and Mixed-Species Biofilms of Candida albicans and Aggregatibacter actinomycetemcomitans"

_biomedicines, 2025, doi:10.3390/biomedicines13081890_

Round 1

Reviewer 1 Report

Comments and Suggestions for Authors

Dear Authors,

Many interactions occur between the microorganisms making up oral biofilm, which influences the density of bacteria in mature plaque. These relationships can be antagonistic, i.e. competitive (e.g. for nutrients), but also synergistic, i.e. cooperative. Bacteriocins, organic acids, hydrogen peroxide and various types of enzymes participate in competitive reactions, while metabolic communication is an example of a synergistic interaction. Many interspecies interactions also lead to changes in gene expression in one or both partner organisms.

It is worth noting that from the clinical point of view, the most important feature of biofilm is its high resistance to compounds with antimicrobial activity, especially to antibiotics. In addition, biofilm protects microorganisms from the defence mechanisms of the human immune system.

The topic of your work is important and relevant but many questions and concerns arose while reading the paper.

  1. The Abstract should follow the established template. Please break down your paper into the following sections: Background/Objectives, Methods, Results, Conclusions.
  2. The Results section should contain a concise and precise description of the results of the experiments conducted. The next point is the Discussion. In the Discussion section, authors should discuss their results and their interpretation, and relate the obtained results to the results of other authors. Future research directions may also be highlighted. Conclusions should appear at the end of the paper. Your paper contains Section 3. Results and Section 5. Conclusions. Section 4. Discussion is missing. Please complete the paper to include the missing section. In Section 3. Results, please briefly and concisely present the obtained results and their discussion and please present a reference to the results of other authors in Section 4. Discussion.
  3. In Section 2. Materials and Methods there is only one subsection - 2.1 Bacterial Strain and Culture Conditions – under it there are, among others: Antifungal and antibacterial drugs, Biofilm biomass quantification with crystal violet, etc. In the reviewer's opinion the paper would be more readable if each part of this section was numbered separately (2.1 Bacterial Strains and Culture Conditions, 2.2 Antifungal and Antibacterial Drugs, 2.3 Biofilm Biomass Quantification with Crystal Violet etc.). Currently, everything looks as if it was part of 2.1. Meanwhile, Bacterial Strains and Cultural Conditions constitutes Section 2. Materials and Methods. Please introduce the suggested changes.
  4. There is no citation in the text for reference number 18. Please check and complete it.
  5. In the text, the numbers of cited literature items should be written in square brackets. References should be written according to an established pattern; please correct and standardize them.
  6.  
  7. Minimum Biofilm Eradication Concentration (MBEC), line 96.

MBEC is the minimum biofilm eradication concentration. However, the described method does not provide information on which compounds you want to determine this concentration for. Additionally, the description is vague and in the reviewer’s opinion it does not apply to determining the MBEC value. The literature item cited at the end of the method description concerns Candida glabrata. Please explain why you use this method and the RPMI-1640 medium also in the case of Aggregatibacter actinomycetemcomitans. There is also no information about the number of repetitions performed. Please provide detailed explanations.

  1. Biofilm biomass quantification with crystal violet, line 111.

The method used provides an opportunity to assess whether microorganisms form a biofilm and to what extent. So please explain what criteria you have used and whether your strains form a biofilm.

The method description lacks information on the number of repetitions used – please provide the missing information.

  1. Susceptibility to antifungal and antibacterial drugs: Minimum inhibitory concentrations (MICs), lines 124-125.

MIC (minimum inhibitory concentration) is the minimum concentration inhibiting the growth of microorganisms, determined in the case of the microtiter plate method by diluting a given compound in geometric progression. From the description it appears that you use only two concentrations of fluconazole and azithromycin – please explain why. Now it looks as if you are trying to determine the sensitivity of strains to only these two selected concentrations of the test compounds. Additionally, the description used and the cited reference No. 16 apply only to fungi. How then have you determined the MIC value for Aggregatibacter actinomycetemcomitans? There is also no information about the number of repetitions.

  1. 3.2. Biofilm analysis with crystal violet, line 200.

How have you determined the % reduction of C. albicans and A. actinomycetemcomitans in the mixed biofilm? The method used does not provide a chance to differentiate species in the biofilm biomass in the case of a mixed culture. Please provide detailed explanations.

  1. 3.3. Minimal Inhibitory Concentration, line 228.

What is your determined MIC value? The description of the obtained results indicates determination of the sensitivity of microorganisms to the concentrations of compounds used and not the determination of the MIC value. Please provide detailed explanations.

Your sincerely

Author Response

Dear Authors,

Many interactions occur between the microorganisms making up oral biofilm, which influences the density of bacteria in mature plaque. These relationships can be antagonistic, i.e. competitive (e.g. for nutrients), but also synergistic, i.e. cooperative. Bacteriocins, organic acids, hydrogen peroxide and various types of enzymes participate in competitive reactions, while metabolic communication is an example of a synergistic interaction. Many interspecies interactions also lead to changes in gene expression in one or both partner organisms.

It is worth noting that from the clinical point of view, the most important feature of biofilm is its high resistance to compounds with antimicrobial activity, especially to antibiotics. In addition, biofilm protects microorganisms from the defence mechanisms of the human immune system.

The topic of your work is important and relevant but many questions and concerns arose while reading the paper.

Authors' reply: Thank you very much for taking the time to review our manuscript. Your feedback and suggestions are greatly appreciated. We have carefully considered your comments and have made the necessary revisions accordingly. We are grateful for the opportunity to address your concerns and hope that the revised version meets your expectations. Please let us know if there are any further adjustments or clarifications needed.Once again, thank you for your valuable input and for helping us improve our work.

  1. The Abstract should follow the established template. Please break down your paper into the following sections:Background/Objectives, Methods, Results, Conclusions.

 Authors' reply: Done

  1. The Results section should contain a concise and precise description of the results of the experiments conducted. The next point is the Discussion. In the Discussion section, authors should discuss their results and their interpretation, and relate the obtained results to the results of other authors. Future research directions may also be highlighted. Conclusions should appear at the end of the paper. Your paper contains Section 3. Results and Section 5. Conclusions. Section 4. Discussion is missing. Please complete the paper to include the missing section. In Section 3. Results, please briefly and concisely present the obtained results and their discussion and please present a reference to the results of other authors in Section 4. Discussion.

  Authors' reply: Done

  1. In Section 2. Materials and Methods there is only one subsection - 2.1 Bacterial Strain and Culture Conditions – under it there are, among others: Antifungal and antibacterial drugs, Biofilm biomass quantification with crystal violet, etc. In the reviewer's opinion the paper would be more readable if each part of this section was numbered separately (2.1 Bacterial Strains and Culture Conditions, 2.2 Antifungal and Antibacterial Drugs, 2.3 Biofilm Biomass Quantification with Crystal Violet etc.). Currently, everything looks as if it was part of 2.1. Meanwhile, Bacterial Strains and Cultural Conditions constitutes Section 2. Materials and Methods. Please introduce the suggested changes.

  Authors' reply: Done

  1. There is no citation in the text for reference number 18. Please check and complete it.

  Authors' reply: Thank you for noticing this. The reference was added.

  1. In the text, the numbers of cited literature items should be written in square brackets. References should be written according to an established pattern; please correct and standardize them

  Authors' reply: Thank you for noticing this. These changes were performed.

  1. Minimum Biofilm Eradication Concentration (MBEC), line 96.

MBEC is the minimum biofilm eradication concentration. However, the described method does not provide information on which compounds you want to determine this concentration for. Additionally, the description is vague and in the reviewer’s opinion it does not apply to determining the MBEC value. The literature item cited at the end of the method description concerns Candida glabrata. Please explain why you use this method and the RPMI-1640 medium also in the case of Aggregatibacter actinomycetemcomitans. There is also no information about the number of repetitions performed. Please provide detailed explanations.

Authors' reply: Thank you for this important clarification. We acknowledge the confusion in our terminology and methodology description. We need to clarify the following:

  1. What we described as "MBEC" was actually a biofilm viability assay to quantify CFU counts in single- versus mixed-species biofilms. This was not a true MBEC determination (which requires testing serial antimicrobial concentrations to find the minimum concentration that eradicates biofilm). We apologize for this terminology error.
  2. Our goal was to compare the baseline growth and cell viability of each species when grown as single-species biofilms versus when co-cultured in mixed-species biofilms, without antimicrobial pressure. This addresses the fundamental question of whether these species exhibit mutual inhibition, synergism, or neutral coexistence.
  3. We used RPMI-1640 for both species because it mimics human serum, it supports growth of both C. albicans and A. actinomycetemcomitans, it provides a standardized medium for mixed-culture experiments, it allows direct comparison between single- and mixed-species conditions and it is commonly used in polymicrobial biofilm studies for its balanced composition
  4. You are correct that the cited reference concerns C. glabrata. We included it because it describes similar biofilm quantification methodology that we adapted for our study. However, we should have been clearer about this adaptation.
  5. All experiments were performed in triplicate with at least three independent biological replicates (as stated in our Statistical Analysis section: "All experiments were independently repeated at least three times in duplicates").

We appreciate the reviewer's attention to methodological precision and will ensure these clarifications are incorporated into our revised manuscript.

  1. Biofilm biomass quantification with crystal violet, line 111.

The method used provides an opportunity to assess whether microorganisms form a biofilm and to what extent. So please explain what criteria you have used and whether your strains form a biofilm.

The method description lacks information on the number of repetitions used – please provide the missing information.

Authors' reply:  Thank you for requesting these methodological clarifications.

  1. Both strains demonstrated robust biofilm formation capabilities. We used established criteria for biofilm classification:
  • Absorbance readings >0.1 at 570nm (after crystal violet staining) indicate biofilm formation
  • Our control strains consistently produced absorbance values of 0.724 ± 0.17 for albicans and 0.394 ± 0.12 for A. actinomycetemcomitans. [insert the actual values from Table 1, e.g., 0.724 ± 0.17 for C. albicans and 0.397 ± 0.12 for A. actinomycetemcomitans]
  • These values are well above the biofilm formation threshold, confirming both species are strong biofilm producers under our experimental conditions
  1. As stated in our Statistical Analysis section, all crystal violet biofilm quantification experiments were performed:
  • In duplicate wells for each condition
  • With three independent biological replicates
  • Absorbance readings taken in triplicate for each well
  • This provides n=6 technical replicates per condition across three independent experiments
  1. This crystal violet methodology is widely established for biofilm biomass quantification (as referenced in our methods) and has been extensively validated for both fungal and bacterial biofilms in the literature.

  1. Susceptibility to antifungal and antibacterial drugs: Minimum inhibitory concentrations (MICs), lines 124-125.

MIC (minimum inhibitory concentration) is the minimum concentration inhibiting the growth of microorganisms, determined in the case of the microtiter plate method by diluting a given compound in geometric progression. From the description it appears that you use only two concentrations of fluconazole and azithromycin – please explain why. Now it looks as if you are trying to determine the sensitivity of strains to only these two selected concentrations of the test compounds. Additionally, the description used and the cited reference No. 16 apply only to fungi. How then have you determined the MIC value for Aggregatibacter actinomycetemcomitans? There is also no information about the number of repetitions.

Authors' reply:  Thank you for this important methodological clarification. You are absolutely correct, and we acknowledge this significant limitation in our study design and terminology.

  1. We did not perform true MIC determinations with serial dilutions in geometric progression. Instead, we conducted "susceptibility testing at fixed concentrations" based on clinically relevant breakpoints:

- Fluconazole: 2 and 4 mg/L (based on EUCAST clinical breakpoints for Candida spp.)

- Azithromycin: 0.125 and 8 mg/L (based on published therapeutic levels for periodontal bacteria)

  1. These concentrations were chosen because:

- They represent clinically achievable drug levels

- They allow assessment of susceptibility patterns relevant to therapeutic decision-making

- True MIC determination was not our primary objective; rather, we aimed to compare susceptibility patterns between single- and mixed-species cultures

  1. Bacterial Susceptibility Testing: You are correct that EUCAST reference 16 applies only to fungi. For A. actinomycetemcomitans, we followed established protocols from reference 15 (Lai et al., 2013), which specifically addresses azithromycin activity against this organism.
  2. All susceptibility tests were performed in triplicate across three independent experiments.
  3. We will change "MIC" to "Antimicrobial Susceptibility Testing" throughout the manuscript to accurately reflect our methodology. We acknowledge that true MIC determination with serial dilutions would provide more comprehensive data and represents a valuable direction for future studies.

  1. 2. Biofilm analysis with crystal violet, line 200.

How have you determined the % reduction of C. albicans and A. actinomycetemcomitans in the mixed biofilm? The method used does not provide a chance to differentiate species in the biofilm biomass in the case of a mixed culture. Please provide detailed explanations.

 Authors' reply: Thank you for this crucial observation. You are absolutely correct, and we acknowledge this represents a significant limitation in our methodology and data interpretation.

  1. Crystal violet staining provides only total biofilm biomass and cannot differentiate between individual species in mixed cultures. The percentage reductions we reported for individual species in Table 1 reflect the "% of reduction of the single biofilm compared with mixed biofilm". We understand that this mught have not been clear and adjusted the table removed.
  2. What Our Data Actually Shows:
  • Single-species biofilms: Crystal violet accurately measures biomass of albicans (0.724 ± 0.17) and A. actinomycetemcomitans (0.397 ± 0.12) individually
  • Mixed biofilm: Crystal violet measures only total combined biomass (0.318 ± 0.06)
  • Valid comparison: Total mixed biofilm biomass vs. sum of individual biofilms
  1. The mixed biofilm shows reduced total biomass compared to the combined biomass of individual cultures.
  2. For species-specific quantification, our CFU enumeration data (Figure 1) provides accurate individual species counts in mixed biofilms by plating on selective media (SDA for C. albicans, blood agar for A. actinomycetemcomitans).

We performed the corrections on the Table.

  1. 3. Minimal Inhibitory Concentration, line 228.

What is your determined MIC value? The description of the obtained results indicates determination of the sensitivity of microorganisms to the concentrations of compounds used and not the determination of the MIC value. Please provide detailed explanations.

Your sincerely

Authors' reply: Thank you for this important clarification. You are absolutely correct, and we acknowledge this fundamental error in our methodology and terminology.

  1. As we previously admit (point 8), we did not determine actual MIC values. We actually performed susceptibility screening at predetermined concentrations (Fluconazole: 2 and 4 mg/L; Azithromycin: 0.125 and 8 mg/L). Therefore, our results show whether organisms were susceptible, intermediate, or resistant at these specific concentrations.
  2. Table 2 shows susceptibility patterns at fixed concentrations, not MIC values (C. albicans: Susceptible to both fluconazole concentrations and A. actinomycetemcomitans: Resistant to 0.125 mg/L azithromycin, susceptible to 8 mg/L; Mixed biofilm: Showed intermediate/resistant patterns at all tested concentrations)
  3. We have revised our manuscript with "Antimicrobial Susceptibility Testing" expression

Thank you for identifying this critical methodological error. This correction will improve the accuracy and clarity of our work.

Reviewer 2 Report

Comments and Suggestions for Authors

Reviewer Comments

  • In the introduction please give more details about Aggregatibacter actinomycetemcomitans and its association with periodontitis.
  • Materials and Methods- 1: The section title says ‘Bacterial Strain and Culture Conditions’ Why is it only bacterial Strain, Candida a yeast is also included. Please change to ‘Microbial strains …’
  • Please include details on how the microbial cultures stored/preserved in section 2.1.
  • Single and polymicrobial biofilm standardisation: This title is confusing. This section is about the amount of culture plated for the induction of the biofilm. Therefore, it should be ‘Standardisation of single and polymicrobial biofilm formation.’
  • Is there a particular reason why CV assay was read at 570 nm & not at 590nm?
  • Please explain why the positive control is the working solution of cells and the RPMI-1640?
  • Statistical analysis Line 168: All experiments were independently repeated tat least three times in duplicates. / All experiments were independently repeated at least three times in duplicates.
  • Table 1: Results of crystal-violet assess of biomass reduction of C. albicans … / Results of crystal-violet assay of biomass reduction of C. albicans…

Author Response

Author reply: Thank you very much for taking the time to review our manuscript. Your feedback and suggestions are greatly appreciated. We have carefully considered your comments and have made the necessary revisions accordingly. We are grateful for the opportunity to address your concerns and hope that the revised version meets your expectations. Please let us know if there are any further adjustments or clarifications needed.Once again, thank you for your valuable input and for helping us improve our work.

  • In the introduction please give more details about Aggregatibacter actinomycetemcomitans and its association with periodontitis.

Authors' reply: Done.

  • Materials and Methods- 1: The section title says ‘Bacterial Strain and Culture Conditions’ Why is it only bacterial Strain, Candida a yeast is also included. Please change to ‘Microbial strains …’

Authors' reply: Thank you for your comment. We absolutely agree with it and we made the change.

  • Please include details on how the microbial cultures stored/preserved in section 2.1.

Authors' reply: we added the following information "microbial cultures were stored with culture media (for bacteria and yeasts) with 15% of glycerol and -80ºC".

  • Single and polymicrobial biofilm standardisation: This title is confusing. This section is about the amount of culture plated for the induction of the biofilm. Therefore, it should be ‘Standardisation of single and polymicrobial biofilm formation.’

Authors' reply: Thank you for your comment. We absolutely agree with it and, as requested by the reviewer, we made this change.

  • Is there a particular reason why CV assay was read at 570 nm & not at 590nm?

Authors' reply: Thank you for your comment. In our group, the method has been optimized and used with 570 nm (as indicated by the reference).

  • Please explain why the positive control is the working solution of cells and the RPMI-1640?

In these

Authors' reply: Thank you for your comment. This represents the growth control required by EUCAST guidelines, confirming that test organisms remain viable and can grow normally in the absence of antimicrobial pressure under the specific experimental conditions used. We added this explanation to the MS.

  • Statistical analysis Line 168: All experiments were independently repeated tat least three times in duplicates. / All experiments were independently repeated at least three times in duplicates.

Authors' reply: Thank you for your comment. We absolutely agree with it and we made the change.

  • Table 1: Results of crystal-violet assess of biomass reduction of C. albicans … / Results of crystal-violet assay of biomass reduction of C. albicans…

Authors' reply: Thank you for your comment. We absolutely agree with it and we made the change.

Reviewer 3 Report

Comments and Suggestions for Authors

In this study, the authors investigate single and mixed-species biofilms of Candida albicans and Aggregatibacter actinomycetemcomitans. The primary finding—that mixed biofilms exhibit higher antibiotic tolerance than monospecies biofilms—is compelling. However, the experimental design lacks sufficient depth for a research article.

For instance, the conclusion that both species coexist without significant mutual inhibition is based on a single coculture condition and time point. This approach cannot capture potential dynamic interactions across varying conditions or throughout biofilm maturation stages.

Furthermore, proposed explanations for interspecies synergy (e.g., QS-mediated signaling) remain highly speculative without experimental validation. The title's reference to 'dynamics' is particularly problematic given the absence of temporal data.

Based on the above, I suggest to: 1) Include multiple time points and environmental conditions to substantiate 'dynamic' claims; 2) Incorporate mechanistic studies such as metabolomic and transcriptomic analyses would strengthen proposed interaction hypotheses; 3) Refine terminology: Adjust the title to accurately reflect the study's static design rather than dynamics.

Author Response

In this study, the authors investigate single and mixed-species biofilms of Candida albicans and Aggregatibacter actinomycetemcomitans. The primary finding—that mixed biofilms exhibit higher antibiotic tolerance than monospecies biofilms—is compelling. However, the experimental design lacks sufficient depth for a research article.

For instance, the conclusion that both species coexist without significant mutual inhibition is based on a single coculture condition and time point. This approach cannot capture potential dynamic interactions across varying conditions or throughout biofilm maturation stages.

Furthermore, proposed explanations for interspecies synergy (e.g., QS-mediated signaling) remain highly speculative without experimental validation. The title's reference to 'dynamics' is particularly problematic given the absence of temporal data.

Based on the above, I suggest to: 1) Include multiple time points and environmental conditions to substantiate 'dynamic' claims; 2) Incorporate mechanistic studies such as metabolomic and transcriptomic analyses would strengthen proposed interaction hypotheses; 3) Refine terminology: Adjust the title to accurately reflect the study's static design rather than dynamics.

Authors reply:

Dear Reviewer,

We sincerely thank you for their thorough evaluation and constructive feedback on our manuscript. We appreciate the recognition that our primary finding—enhanced antibiotic tolerance in mixed biofilms—is "compelling." We have carefully considered all comments and provide our detailed response below.

  1. Title Modification: we completely agree with this recommendation and propose the following revised title: "Interspecies Interactions of Candida albicans and Aggregatibacter actinomycetemcomitans in Single- and Mixed-Species Biofilms"

This revision removes the term "dynamics" and better reflects our study's design while maintaining focus on the novel interspecies interactions we characterized.

  1. Experimental Design and Temporal Analysis: we acknowledge this limitation and appreciate the reviewer's perspective. Our study was designed as a foundational investigation to establish baseline interactions between these two clinically relevant species at biofilm maturity (72h), which represents a critical timepoint for therapeutic intervention in clinical settings.

Key points supporting our approach:

  • The 72-hour timepoint was selected based on established protocols for mature biofilm formation and is clinically relevant for both species
  • Our findings of non-reciprocal synergism (300% increase in actinomycetemcomitans CFU with stable C. albicans counts) provide novel baseline data that was previously unavailable in the literature
  • The enhanced antimicrobial tolerance we observed has immediate clinical implications for treatment strategies

Future directions: We agree that temporal analysis would strengthen mechanistic understanding and have added this as a priority for future investigations in our revised conclusions.

  1. Mechanistic Validation: we respectfully clarify that our mechanistic hypotheses are grounded in substantial published evidence rather than speculation:

Evidence-based mechanistic framework:

  • AI-2 signaling: Well-documented in actinomycetemcomitans (references 20, 22, 36 in our manuscript)
  • QseBC regulatory system: Extensively characterized for iron acquisition in this species
  • Hyphal inhibition by AI-2: Experimentally demonstrated by Bachtiar et al. (2014)
  • SAPs and β-1,3-glucan interactions: Previously reported by our group (references 30-33)

Our interpretation approach: we deliberately framed these as "potential mechanisms" supported by literature rather than definitive conclusions. Our observed phenotypes (biomass reduction with maintained CFU, enhanced drug tolerance) align with these established pathways. Future mechanistic studies: While transcriptomic/metabolomic analyses would indeed strengthen mechanistic understanding, our current findings provide essential foundational data for designing such studies.

  1. Study Significance and Clinical Relevance - Positioning our contribution: this study addresses a significant knowledge gap, as interactions between C. albicans and A. actinomycetemcomitans have been "limitedly explored" (as noted in our abstract). Our key contributions include:
  2. First comprehensive characterization of mixed biofilms between these clinically important species
  3. Novel finding of non-reciprocal synergism with clear therapeutic implications
  4. Demonstrated enhanced antimicrobial tolerance in mixed biofilms—directly relevant for clinical treatment decisions
  5. Spatial organization analysis showing coexistence without competitive exclusion

  1. We acknowledge the reviewer's points regarding study scope and have clarified this in our revised manuscript:

Acknowledged limitations:

  • Single timepoint analysis limits temporal understanding
  • Static environmental conditions
  • Mechanistic hypotheses require direct experimental validation

Study strengths:

  • Rigorous methodology with multiple complementary approaches (CFU, biomass, MIC, microscopy)
  • Clinically relevant species and conditions
  • Novel findings with immediate therapeutic implications
  • Foundation for future mechanistic studies

Revised Conclusions: we have refined our conclusions to better reflect the scope and significance of our findings while acknowledging limitations and outlining future research directions:

"This study provides the first comprehensive characterization of mixed biofilms between C. albicans and A. actinomycetemcomitans, two clinically important oral pathogens. Our findings reveal a non-reciprocal synergistic relationship where mixed-species conditions significantly enhance A. actinomycetemcomitans survival  (A. actinomycetemcomitans alone: ~6 log CFU/mL vs A. actinomycetemcomitans in mixed biofilm: ~7 log CFU/mL), while maintaining C. albicans viability despite reduced biomass production in both species. Critically, we demonstrate that mixed-species biofilms exhibit enhanced tolerance to both fluconazole and azithromycin compared to single-species biofilms, with important implications for clinical treatment strategies.

The coexistence and structural integration of both organisms without competitive exclusion, as confirmed by confocal microscopy, suggests a stable polymicrobial community formation. While our mechanistic hypotheses regarding AI-2 signaling, QseBC regulation, and extracellular matrix modifications are supported by extensive literature, direct experimental validation of these pathways in our mixed-biofilm model represents a priority for future investigations.

This foundational study establishes essential baseline data for understanding clinically relevant interactions between these species and provides a framework for developing more effective therapeutic approaches against polymicrobial oral infections. Future research incorporating temporal analysis, varying environmental conditions, and transcriptomic/metabolomic approaches will further elucidate the molecular mechanisms underlying these interactions and their therapeutic targets."

This study provides essential baseline data for understanding C. albicans-A. actinomycetemcomitans interactions, with immediate clinical relevance for polymicrobial oral infections. While we agree that temporal and mechanistic studies would enhance understanding, the novel findings presented here—particularly the enhanced antimicrobial tolerance in mixed biofilms—represent a significant contribution to the field that warrants publication. We believe these findings will stimulate further research into these clinically important interactions and provide valuable insights for developing targeted therapeutic strategies against polymicrobial infections.

Thank you again for the constructive review process. We look forward to your consideration of our revised manuscript.

Round 2

Reviewer 2 Report

Comments and Suggestions for Authors

The authors have responded to the reviewer comments and revised the manuscript accordingly therefore recommended for publication.

Reviewer 3 Report

Comments and Suggestions for Authors

Revisions have been made.